# An Accelerated Proximal Coordinate Gradient Method

**Qihang Lin**
University of Iowa
Iowa City, IA, USA
qihang-lin@uiowa.edu

**Zhaosong Lu**
Simon Fraser University
Burnaby, BC, Canada
zhaosong@sfu.ca

**Lin Xiao**
Microsoft Research
Redmond, WA, USA
lin.xiao@microsoft.com

## Abstract

We develop an accelerated randomized proximal coordinate gradient (APCG) method, for solving a broad class of composite convex optimization problems. In particular, our method achieves faster linear convergence rates for minimizing strongly convex functions than existing randomized proximal coordinate gradient methods. We show how to apply the APCG method to solve the dual of the regularized empirical risk minimization (ERM) problem, and devise efficient implementations that avoid full-dimensional vector operations. For ill-conditioned ERM problems, our method obtains improved convergence rates than the state-of-the-art stochastic dual coordinate ascent (SDCA) method.

## 1 Introduction

Coordinate descent methods have received extensive attention in recent years due to their potential for solving large-scale optimization problems arising from machine learning and other applications. In this paper, we develop an accelerated proximal coordinate gradient (APCG) method for solving convex optimization problems with the following form:

$$\underset{x \in \mathbb{R}^N}{\text{minimize}} \quad \{F(x) \overset{\text{def}}{=} f(x) + \Psi(x)\}, \tag{1}$$

where $f$ is differentiable on $\text{dom}(\Psi)$, and $\Psi$ has a block separable structure. More specifically,

$$\Psi(x) = \sum_{i=1}^{n} \Psi_i(x_i), \tag{2}$$

where each $x_i$ denotes a sub-vector of $x$ with cardinality $N_i$, and each $\Psi_i : \mathbb{R}^{N_i} \to \mathbb{R} \cup \{+\infty\}$ is a closed convex function. We assume the collection $\{x_i : i = 1, \ldots, n\}$ form a partition of the components of $x \in \mathbb{R}^N$. In addition to the capability of modeling nonsmooth regularization terms such as $\Psi(x) = \lambda \|x\|_1$, this model also includes optimization problems with block separable constraints. More precisely, each block constraint $x_i \in C_i$, where $C_i$ is a closed convex set, can be modeled by an indicator function defined as $\Psi_i(x_i) = 0$ if $x_i \in C_i$ and $\infty$ otherwise.

At each iteration, coordinate descent methods choose one block of coordinates $x_i$ to sufficiently reduce the objective value while keeping other blocks fixed. One common and simple approach for choosing such a block is the *cyclic* scheme. The global and local convergence properties of the cyclic coordinate descent method have been studied in, for example, [21, 11, 16, 2, 5]. Recently, *randomized* strategies for choosing the block to update became more popular. In addition to its theoretical benefits [13, 14, 19], numerous experiments have demonstrated that randomized coordinate descent methods are very powerful for solving large-scale machine learning problems [3, 6, 18, 19].

Inspired by the success of accelerated full gradient methods (e.g., [12, 1, 22]), several recent work applied Nesterov's acceleration schemes to speed up randomized coordinate descent methods. In particular, Nesterov [13] developed an accelerated randomized coordinate gradient method for minimizing unconstrained smooth convex functions, which corresponds to the case of $\Psi(x) \equiv 0$ in (1).

Lu and Xiao [10] gave a sharper convergence analysis of Nesterov's method, and Lee and Sidford [8] developed extensions with weighted random sampling schemes. More recently, Fercoq and Richtárik [4] proposed an APPROX (Accelerated, Parallel and PROXimal) coordinate descent method for solving the more general problem (1) and obtained accelerated sublinear convergence rates, but their method cannot exploit the strong convexity to obtain accelerated linear rates.

In this paper, we develop a general APCG method that achieves accelerated linear convergence rates when the objective function is strongly convex. Without the strong convexity assumption, our method recovers the APPROX method [4]. Moreover, we show how to apply the APCG method to solve the dual of the regularized empirical risk minimization (ERM) problem, and devise efficient implementations that avoid full-dimensional vector operations. For ill-conditioned ERM problems, our method obtains faster convergence rates than the state-of-the-art stochastic dual coordinate ascent (SDCA) method [19], and the improved iteration complexity matches the accelerated SDCA method [20]. We present numerical experiments to illustrate the advantage of our method.

## 1.1 Notations and assumptions

For any partition of $x \in \mathbb{R}^N$ into $\{x_i \in \mathbb{R}^{N_i} : i = 1, \ldots, n\}$, there is an $N \times N$ permutation matrix $U$ partitioned as $U = [U_1 \cdots U_n]$, where $U_i \in \mathbb{R}^{N \times N_i}$, such that

$$x = \sum_{i=1}^{n} U_i x_i, \qquad \text{and} \quad x_i = U_i^T x, \quad i = 1, \ldots, n.$$

For any $x \in \mathbb{R}^N$, the *partial gradient* of $f$ with respect to $x_i$ is defined as

$$\nabla_i f(x) = U_i^T \nabla f(x), \quad i = 1, \ldots, n.$$

We associate each subspace $\mathbb{R}^{N_i}$, for $i = 1, \ldots, n$, with the standard Euclidean norm, denoted by $\| \cdot \|$. We make the following assumptions which are standard in the literature on coordinate descent methods (e.g., [13, 14]).

**Assumption 1.** *The gradient of function $f$ is block-wise Lipschitz continuous with constants $L_i$, i.e.,*

$$\|\nabla_i f(x + U_i h_i) - \nabla_i f(x)\| \leq L_i \|h_i\|, \quad \forall h_i \in \mathbb{R}^{N_i}, \quad i = 1, \ldots, n, \quad x \in \mathbb{R}^N.$$

For convenience, we define the following norm in the whole space $\mathbb{R}^N$:

$$\|x\|_L \quad = \quad \left( \sum_{i=1}^{n} L_i \|x_i\|^2 \right)^{1/2}, \quad \forall x \in \mathbb{R}^N. \tag{3}$$

**Assumption 2.** *There exists $\mu \geq 0$ such that for all $y \in \mathbb{R}^N$ and $x \in \mathrm{dom}\,(\Psi)$,*

$$f(y) \geq f(x) + \langle \nabla f(x), y - x \rangle + \frac{\mu}{2} \|y - x\|_L^2.$$

The convexity parameter of $f$ with respect to the norm $\| \cdot \|_L$ is the largest $\mu$ such that the above inequality holds. Every convex function satisfies Assumption 2 with $\mu = 0$. If $\mu > 0$, the function $f$ is called *strongly* convex.

We note that an immediate consequence of Assumption 1 is

$$f(x + U_i h_i) \leq f(x) + \langle \nabla_i f(x), h_i \rangle + \frac{L_i}{2} \|h_i\|^2, \quad \forall h_i \in \mathbb{R}^{N_i}, \quad i = 1, \ldots, n, \quad x \in \mathbb{R}^N. \tag{4}$$

This together with Assumption 2 implies $\mu \leq 1$.

## 2 The APCG method

In this section we describe the general APCG method, and several variants that are more suitable for implementation under different assumptions. These algorithms extend Nesterov's accelerated gradient methods [12, Section 2.2] to the composite and coordinate descent setting.

We first explain the notations used in our algorithms. The algorithms proceed in iterations, with $k$ being the iteration counter. Lower case letters $x$, $y$, $z$ represent vectors in the full space $\mathbb{R}^N$, and $x^{(k)}$, $y^{(k)}$ and $z^{(k)}$ are their values at the $k$th iteration. Each block coordinate is indicated with a subscript, for example, $x_i^{(k)}$ represents the value of the $i$th block of the vector $x^{(k)}$. The Greek letters $\alpha$, $\beta$, $\gamma$ are scalars, and $\alpha_k$, $\beta_k$ and $\gamma_k$ represent their values at iteration $k$.

---

**Algorithm 1:** the APCG method

**Input:** $x^{(0)} \in \mathrm{dom}\,(\Psi)$ and convexity parameter $\mu \geq 0$.

**Initialize:** set $z^{(0)} = x^{(0)}$ and choose $0 < \gamma_0 \in [\mu, 1]$.

**Iterate:** repeat for $k = 0, 1, 2, \ldots$

   1. Compute $\alpha_k \in (0, \frac{1}{n}]$ from the equation

$$n^2 \alpha_k^2 = (1 - \alpha_k)\,\gamma_k + \alpha_k \mu, \tag{5}$$

      and set

$$\gamma_{k+1} = (1 - \alpha_k)\gamma_k + \alpha_k \mu, \qquad \beta_k = \frac{\alpha_k \mu}{\gamma_{k+1}}. \tag{6}$$

   2. Compute $y^{(k)}$ as

$$y^{(k)} = \frac{1}{\alpha_k \gamma_k + \gamma_{k+1}}\left(\alpha_k \gamma_k z^{(k)} + \gamma_{k+1} x^{(k)}\right). \tag{7}$$

   3. Choose $i_k \in \{1, \ldots, n\}$ uniformly at random and compute

$$z^{(k+1)} = \arg\min_{x \in \mathbb{R}^N}\left\{\frac{n\alpha_k}{2}\left\|x - (1-\beta_k)z^{(k)} - \beta_k y^{(k)}\right\|_L^2 + \langle \nabla_{i_k} f(y^{(k)}), x_{i_k}\rangle + \Psi_{i_k}(x_{i_k})\right\}.$$

   4. Set

$$x^{(k+1)} = y^{(k)} + n\alpha_k(z^{(k+1)} - z^{(k)}) + \frac{\mu}{n}(z^{(k)} - y^{(k)}). \tag{8}$$

---

The general APCG method is given as Algorithm 1. At each iteration $k$, it chooses a random coordinate $i_k \in \{1, \ldots, n\}$ and generates $y^{(k)}$, $x^{(k+1)}$ and $z^{(k+1)}$. One can observe that $x^{(k+1)}$ and $z^{(k+1)}$ depend on the realization of the random variable

$$\xi_k = \{i_0, i_1, \ldots, i_k\},$$

while $y^{(k)}$ is independent of $i_k$ and only depends on $\xi_{k-1}$. To better understand this method, we make the following observations. For convenience, we define

$$\tilde{z}^{(k+1)} = \arg\min_{x \in \mathbb{R}^N}\left\{\frac{n\alpha_k}{2}\left\|x - (1-\beta_k)z^{(k)} - \beta_k y^{(k)}\right\|_L^2 + \langle \nabla f(y^{(k)}), x - y^{(k)}\rangle + \Psi(x)\right\}, \tag{9}$$

which is a full-dimensional update version of Step 3. One can observe that $z^{(k+1)}$ is updated as

$$z_i^{(k+1)} = \begin{cases} \tilde{z}_i^{(k+1)} & \text{if } i = i_k, \\ (1 - \beta_k)z_i^{(k)} + \beta_k y_i^{(k)} & \text{if } i \neq i_k. \end{cases} \tag{10}$$

Notice that from (5), (6), (7) and (8) we have

$$x^{(k+1)} = y^{(k)} + n\alpha_k\left(z^{(k+1)} - (1 - \beta_k)z^{(k)} - \beta_k y^{(k)}\right),$$

which together with (10) yields

$$x_i^{(k+1)} = \begin{cases} y_i^{(k)} + n\alpha_k\left(z_i^{(k+1)} - z_i^{(k)}\right) + \frac{\mu}{n}\left(z_i^{(k)} - y_i^{(k)}\right) & \text{if } i = i_k, \\ y_i^{(k)} & \text{if } i \neq i_k. \end{cases} \tag{11}$$

That is, in Step 4, we only need to update the block coordinates $x_{i_k}^{(k+1)}$ and set the rest to be $y_i^{(k)}$.

We now state a theorem concerning the expected rate of convergence of the APCG method, whose proof can be found in the full report [9].

**Theorem 1.** *Suppose Assumptions 1 and 2 hold. Let $F^\star$ be the optimal value of problem (1), and $\{x^{(k)}\}$ be the sequence generated by the APCG method. Then, for any $k \geq 0$, there holds:*

$$\mathbf{E}_{\xi_{k-1}}[F(x^{(k)})] - F^\star \leq \min\left\{\left(1 - \frac{\sqrt{\mu}}{n}\right)^k, \left(\frac{2n}{2n + k\sqrt{\gamma_0}}\right)^2\right\}\left(F(x^{(0)}) - F^\star + \frac{\gamma_0}{2}R_0^2\right),$$

*where*

$$R_0 \overset{\text{def}}{=} \min_{x^\star \in X^\star}\|x^{(0)} - x^\star\|_L, \tag{12}$$

*and $X^\star$ is the set of optimal solutions of problem (1).*

Our result in Theorem 1 improves upon the convergence rates of the proximal coordinate gradient methods in [14, 10], which have convergence rates on the order of

$$O\left(\min\left\{\left(1 - \frac{\mu}{n}\right)^k, \ \frac{n}{n+k}\right\}\right).$$

For $n = 1$, our result matches exactly that of the accelerated full gradient method in [12, Section 2.2].

## 2.1 Two special cases

Here we give two simplified versions of the APCG method, for the special cases of $\mu = 0$ and $\mu > 0$, respectively. Algorithm 2 shows the simplified version for $\mu = 0$, which can be applied to problems without strong convexity, or if the convexity parameter $\mu$ is unknown.

---

**Algorithm 2:** APCG with $\mu = 0$

**Input:** $x^{(0)} \in \text{dom}(\Psi)$.

**Initialize:** set $z^{(0)} = x^{(0)}$ and choose $\alpha_0 \in (0, \frac{1}{n}]$.

**Iterate:** repeat for $k = 0, 1, 2, \ldots$

    1. Compute $y^{(k)} = (1 - \alpha_k)x^{(k)} + \alpha_k z^{(k)}$.

    2. Choose $i_k \in \{1, \ldots, n\}$ uniformly at random and compute
$$z_{i_k}^{(k+1)} = \arg\min_{x \in \mathbb{R}^N}\left\{\frac{n\alpha_k L_{i_k}}{2}\left\|x - z_{i_k}^{(k)}\right\|^2 + \langle \nabla_{i_k} f(y^{(k)}), x - y_{i_k}^{(k)}\rangle + \Psi_{i_k}(x)\right\}.$$
      and set $z_i^{(k+1)} = z_i^{(k)}$ for all $i \neq i_k$.

    3. Set $x^{(k+1)} = y^{(k)} + n\alpha_k(z^{(k+1)} - z^{(k)})$.

    4. Compute $\alpha_{k+1} = \frac{1}{2}\left(\sqrt{\alpha_k^4 + 4\alpha_k^2} - \alpha_k^2\right)$.

---

According to Theorem 1, Algorithm 2 has an accelerated sublinear convergence rate, that is

$$\mathbf{E}_{\xi_{k-1}}[F(x^{(k)})] - F^\star \leq \left(\frac{2n}{2n + kn\alpha_0}\right)^2\left(F(x^{(0)}) - F^\star + \frac{1}{2}R_0^2\right).$$

With the choice of $\alpha_0 = 1/n$, Algorithm 2 reduces to the APPROX method [4] with single block update at each iteration (i.e., $\tau = 1$ in their Algorithm 1).

For the strongly convex case with $\mu > 0$, we can initialize Algorithm 1 with the parameter $\gamma_0 = \mu$, which implies $\gamma_k = \mu$ and $\alpha_k = \beta_k = \sqrt{\mu}/n$ for all $k \geq 0$. This results in Algorithm 3.

---

**Algorithm 3:** APCG with $\gamma_0 = \mu > 0$

**Input:** $x^{(0)} \in \text{dom}(\Psi)$ and convexity parameter $\mu > 0$.

**Initialize:** set $z^{(0)} = x^{(0)}$ and and $\alpha = \frac{\sqrt{\mu}}{n}$.

**Iterate:** repeat for $k = 0, 1, 2, \ldots$

    1. Compute $y^{(k)} = \frac{x^{(k)} + \alpha z^{(k)}}{1 + \alpha}$.

    2. Choose $i_k \in \{1, \ldots, n\}$ uniformly at random and compute
$$z^{(k+1)} = \arg\min_{x \in \mathbb{R}^N}\left\{\frac{n\alpha}{2}\left\|x - (1-\alpha)z^{(k)} - \alpha y^{(k)}\right\|_L^2 + \langle \nabla_{i_k} f(y^{(k)}), x_{i_k} - y_{i_k}^{(k)}\rangle + \Psi_{i_k}(x_{i_k})\right\}.$$

    3. Set $x^{(k+1)} = y^{(k)} + n\alpha(z^{(k+1)} - z^{(k)}) + n\alpha^2(z^{(k)} - y^{(k)})$.

---

As a direct corollary of Theorem 1, Algorithm 3 enjoys an accelerated linear convergence rate:

$$\mathbf{E}_{\xi_{k-1}}[F(x^{(k)})] - F^\star \leq \left(1 - \frac{\sqrt{\mu}}{n}\right)^k\left(F(x^{(0)}) - F^\star + \frac{\mu}{2}R_0^2\right).$$

To the best of our knowledge, this is the first time such an accelerated rate is obtained for solving the general problem (1) (with strong convexity) using coordinate descent type of methods.

## 2.2 Efficient implementation

The APCG methods we presented so far all need to perform full-dimensional vector operations at each iteration. For example, $y^{(k)}$ is updated as a convex combination of $x^{(k)}$ and $z^{(k)}$, and this can be very costly since in general they can be dense vectors. Moreover, for the strongly convex case (Algorithms 1 and 3), all blocks of $z^{(k+1)}$ need to be updated at each iteration, although only the $i_k$-th block needs to compute the partial gradient and perform a proximal mapping. These full-dimensional vector updates cost $O(N)$ operations per iteration and may cause the overall computational cost of APCG to be even higher than the full gradient methods (see discussions in [13]).

In order to avoid full-dimensional vector operations, Lee and Sidford [8] proposed a change of variables scheme for accelerated coordinate descent methods for unconstrained smooth minimization. Fercoq and Richtárik [4] devised a similar scheme for efficient implementation in the $\mu = 0$ case for composite minimization. Here we show that such a scheme can also be developed for the case of $\mu > 0$ in the composite optimization setting. For simplicity, we only present an equivalent implementation of the simplified APCG method described in Algorithm 3.

---

**Algorithm 4:** Efficient implementation of APCG with $\gamma_0 = \mu > 0$

**Input:** $x^{(0)} \in \mathrm{dom}\,(\Psi)$ and convexity parameter $\mu > 0$.

**Initialize:** set $\alpha = \frac{\sqrt{\mu}}{n}$ and $\rho = \frac{1-\alpha}{1+\alpha}$, and initialize $u^{(0)} = 0$ and $v^{(0)} = x^{(0)}$.

**Iterate:** repeat for $k = 0, 1, 2, \dots$

  1. Choose $i_k \in \{1, \dots, n\}$ uniformly at random and compute

$$\Delta_{i_k}^{(k)} = \arg\min_{\Delta \in \mathbb{R}^{N_{i_k}}} \left\{ \frac{n\alpha L_{i_k}}{2}\|\Delta\|^2 + \langle \nabla_{i_k} f(\rho^{k+1} u^{(k)} + v^{(k)}), \Delta \rangle + \Psi_{i_k}(-\rho^{k+1} u_{i_k}^{(k)} + v_{i_k}^{(k)} + \Delta) \right\}.$$

  2. Let $u^{(k+1)} = u^{(k)}$ and $v^{(k+1)} = v^{(k)}$, and update

$$u_{i_k}^{(k+1)} = u_{i_k}^{(k)} - \frac{1-n\alpha}{2\rho^{k+1}}\Delta_{i_k}^{(k)}, \qquad v_{i_k}^{(k+1)} = v_{i_k}^{(k)} + \frac{1+n\alpha}{2}\Delta_{i_k}^{(k)}. \tag{13}$$

**Output:** $x^{(k+1)} = \rho^{k+1} u^{(k+1)} + v^{(k+1)}$

---

The following Proposition is proved in the full report [9].

**Proposition 1.** *The iterates of Algorithm 3 and Algorithm 4 satisfy the following relationships:*

$$x^{(k)} = \rho^k u^{(k)} + v^{(k)}, \qquad y^{(k)} = \rho^{k+1} u^{(k)} + v^{(k)}, \qquad z^{(k)} = -\rho^k u^{(k)} + v^{(k)}. \tag{14}$$

We note that in Algorithm 4, only a single block coordinate of the vectors $u^{(k)}$ and $v^{(k)}$ are updated at each iteration, which cost $O(N_i)$. However, computing the partial gradient $\nabla_{i_k} f(\rho^{k+1} u^{(k)} + v^{(k)})$ may still cost $O(N)$ in general. In the next section, we show how to further exploit structure in many ERM problems to completely avoid full-dimensional vector operations.

## 3 Application to regularized empirical risk minimization (ERM)

Let $A_1, \dots, A_n$ be vectors in $\mathbb{R}^d$, $\phi_1, \dots, \phi_n$ be a sequence of convex functions defined on $\mathbb{R}$, and $g$ be a convex function on $\mathbb{R}^d$. Regularized ERM aims to solve the following problem:

$$\underset{w \in \mathbb{R}^d}{\mathrm{minimize}}\ P(w), \quad \text{with} \quad P(w) = \frac{1}{n}\sum_{i=1}^{n}\phi_i(A_i^T w) + \lambda g(w),$$

where $\lambda > 0$ is a regularization parameter. For example, given a label $b_i \in \{\pm 1\}$ for each vector $A_i$, for $i = 1, \dots, n$, we obtain the linear SVM problem by setting $\phi_i(z) = \max\{0, 1-b_i z\}$ and $g(w) = (1/2)\|w\|_2^2$. Regularized logistic regression is obtained by setting $\phi_i(z) = \log(1+\exp(-b_i z))$. This formulation also includes regression problems. For example, ridge regression is obtained by setting $(1/2)\phi_i(z) = (z - b_i)^2$ and $g(w) = (1/2)\|w\|_2^2$, and we get Lasso if $g(w) = \|w\|_1$.

Let $\phi_i^*$ be the convex conjugate of $\phi_i$, that is, $\phi_i^*(u) = \max_{z \in \mathbb{R}}(zu - \phi_i(z))$. The dual of the regularized ERM problem is (see, e.g., [19])

$$\underset{x \in \mathbb{R}^n}{\text{maximize }} D(x), \quad \text{with} \quad D(x) = \frac{1}{n}\sum_{i=1}^{n} -\phi_i^*(-x_i) - \lambda g^*\left(\frac{1}{\lambda n}Ax\right),$$

where $A = [A_1, \ldots, A_n]$. This is equivalent to minimize $F(x) \stackrel{\text{def}}{=} -D(x)$, that is,

$$\underset{x \in \mathbb{R}^n}{\text{minimize }} F(x) \stackrel{\text{def}}{=} \frac{1}{n}\sum_{i=1}^{n} \phi_i^*(-x_i) + \lambda g^*\left(\frac{1}{\lambda n}Ax\right).$$

The structure of $F(x)$ above matches the formulation in (1) and (2) with $f(x) = \lambda g^*\left(\frac{1}{\lambda n}Ax\right)$ and $\Psi_i(x_i) = \frac{1}{n}\phi_i^*(-x_i)$, and we can apply the APCG method to minimize $F(x)$. In order to exploit the fast linear convergence rate, we make the following assumption.

**Assumption 3.** *Each function $\phi_i$ is $1/\gamma$ smooth, and the function $g$ has unit convexity parameter 1.*

Here we slightly abuse the notation by overloading $\gamma$, which also appeared in Algorithm 1. But in this section it solely represents the (inverse) smoothness parameter of $\phi_i$. Assumption 3 implies that each $\phi_i^*$ has strong convexity parameter $\gamma$ (with respect to the local Euclidean norm) and $g^*$ is differentiable and $\nabla g^*$ has Lipschitz constant 1. In the following, we split the function $F(x) = f(x) + \Psi(x)$ by relocating the strong convexity term as follows:

$$f(x) = \lambda g^*\left(\frac{1}{\lambda n}Ax\right) + \frac{\gamma}{2n}\|x\|^2, \qquad \Psi(x) = \frac{1}{n}\sum_{i=1}^{n}\left(\phi^*(-x_i) - \frac{\gamma}{2}\|x_i\|^2\right). \tag{15}$$

As a result, the function $f$ is strongly convex and each $\Psi_i$ is still convex. Now we can apply the APCG method to minimize $F(x) = -D(x)$, and obtain the following guarantee.

**Theorem 2.** *Suppose Assumption 3 holds and $\|A_i\| \leq R$ for all $i = 1, \ldots, n$. In order to obtain an expected dual optimality gap $\mathbf{E}[D^\star - D(x^{(k)})] \leq \epsilon$ by using the APCG method, it suffices to have*

$$k \geq \left(n + \sqrt{\frac{nR^2}{\lambda \gamma}}\right)\log(C/\epsilon). \tag{16}$$

*where $D^\star = \max_{x \in \mathbb{R}^n} D(x)$ and the constant $C = D^\star - D(x^{(0)}) + (\gamma/(2n))\|x^{(0)} - x^\star\|^2$.*

*Proof.* The function $f(x)$ in (15) has coordinate Lipschitz constants $L_i = \frac{\|A_i\|^2}{\lambda n^2} + \frac{\gamma}{n} \leq \frac{R^2 + \lambda \gamma n}{\lambda n^2}$ and convexity parameter $\frac{\gamma}{n}$ with respect to the unweighted Euclidean norm. The strong convexity parameter of $f(x)$ with respect to the norm $\|\cdot\|_L$ defined in (3) is

$$\mu = \frac{\gamma}{n}\bigg/\frac{R^2 + \lambda \gamma n}{\lambda n^2} = \frac{\lambda \gamma n}{R^2 + \lambda \gamma n}.$$

According to Theorem 1, we have $\mathbf{E}[D^\star - D(x^{(0)})] \leq \left(1 - \frac{\sqrt{\mu}}{n}\right)^k C \leq \exp\left(-\frac{\sqrt{\mu}}{n}k\right)C$. Therefore it suffices to have the number of iterations $k$ to be larger than

$$\frac{n}{\sqrt{\mu}}\log(C/\epsilon) = n\sqrt{\frac{R^2 + \lambda \gamma n}{\lambda \gamma n}}\log(C/\epsilon) = \sqrt{n^2 + \frac{nR^2}{\lambda \gamma}}\log(C/\epsilon) \leq \left(n + \sqrt{\frac{nR^2}{\lambda \gamma}}\right)\log(C/\epsilon).$$

This finishes the proof. $\square$

Several state-of-the-art algorithms for ERM, including SDCA [19], SAG [15, 17] and SVRG [7, 23] obtain the iteration complexity

$$O\left(\left(n + \frac{R^2}{\lambda \gamma}\right)\log(1/\epsilon)\right). \tag{17}$$

We note that our result in (16) can be much better for ill-conditioned problems, i.e., when the condition number $\frac{R^2}{\lambda \gamma}$ is larger than $n$. This is also confirmed by our numerical experiments in Section 4.

The complexity bound in (17) for the aforementioned work is for minimizing the primal objective $P(x)$ or the duality gap $P(x) - D(x)$, but our result in Theorem 2 is in terms of the dual optimality. In the full report [9], we show that the same guarantee on accelerated primal-dual convergence can be obtained by our method with an extra primal gradient step, without affecting the overall complexity. The experiments in Section 4 illustrate superior performance of our algorithm on reducing the primal objective value, even without performing the extra step.

We note that Shalev-Shwartz and Zhang [20] recently developed an accelerated SDCA method which achieves the same complexity $O\left(\left(n + \sqrt{\frac{n}{\lambda\gamma}}\right)\log(1/\epsilon)\right)$ as our method. Their method calls the SDCA method in a full-dimensional accelerated gradient method in an inner-outer iteration procedure. In contrast, our APCG method is a straightforward single loop coordinate gradient method.

### 3.1 Implementation details

Here we show how to exploit the structure of the regularized ERM problem to efficiently compute the coordinate gradient $\nabla_{i_k} f(y^{(k)})$, and totally avoid full-dimensional updates in Algorithm 4. We focus on the special case $g(w) = \frac{1}{2}\|w\|^2$ and show how to compute $\nabla_{i_k} f(y^{(k)})$. According to (15),

$$\nabla_{i_k} f(y^{(k)}) = \frac{1}{\lambda n^2} A_i^T (A y^{(k)}) + \frac{\gamma}{n} y_{i_k}^{(k)}.$$

Since we do not form $y^{(k)}$ in Algorithm 4, we update $Ay^{(k)}$ by storing and updating two vectors in $\mathbb{R}^d$: $p^{(k)} = Au^{(k)}$ and $q^{(k)} = Av^{(k)}$. The resulting method is detailed in Algorithm 5.

---

**Algorithm 5:** APCG for solving dual ERM

**Input:** $x^{(0)} \in \text{dom}(\Psi)$ and convexity parameter $\mu > 0$.

**Initialize:** set $\alpha = \frac{\sqrt{\mu}}{n}$ and $\rho = \frac{1-\alpha}{1+\alpha}$, and let $u^{(0)} = 0$, $v^{(0)} = x^{(0)}$, $p^{(0)} = 0$ and $q^{(0)} = Ax^{(0)}$.

**Iterate:** repeat for $k = 0, 1, 2, \dots$

    1. Choose $i_k \in \{1, \dots, n\}$ uniformly at random, compute the coordinate gradient

$$\nabla_{i_k}^{(k)} = \frac{1}{\lambda n^2}\left(\rho^{k+1} A_{i_k}^T p^{(k)} + A_{i_k}^T q^{(k)}\right) + \frac{\gamma}{n}\left(\rho^{k+1} u_{i_k}^{(k)} + v_{i_k}^{(k)}\right).$$

    2. Compute coordinate increment

$$\Delta_{i_k}^{(k)} = \underset{\Delta \in \mathbb{R}^{N_{i_k}}}{\arg\min}\left\{\frac{\alpha(\|A_{i_k}\|^2 + \lambda\gamma n)}{2\lambda n}\|\Delta\|^2 + \langle\nabla_{i_k}^{(k)}, \Delta\rangle + \frac{1}{n}\phi_{i_k}^*\left(\rho^{k+1} u_{i_k}^{(k)} - v_{i_k}^{(k)} - \Delta\right)\right\}.$$

    3. Let $u^{(k+1)} = u^{(k)}$ and $v^{(k+1)} = v^{(k)}$, and update

$$u_{i_k}^{(k+1)} = u_{i_k}^{(k)} - \frac{1-n\alpha}{2\rho^{k+1}}\Delta_{i_k}^{(k)}, \qquad\qquad v_{i_k}^{(k+1)} = v_{i_k}^{(k)} + \frac{1+n\alpha}{2}\Delta_{i_k}^{(k)},$$

$$p^{(k+1)} = p^{(k)} - \frac{1-n\alpha}{2\rho^{k+1}}A_{i_k}\Delta_{i_k}^{(k)}, \qquad\qquad q^{(k+1)} = q^{(k)} + \frac{1+n\alpha}{2}A_{i_k}\Delta_{i_k}^{(k)}. \quad (18)$$

**Output:** approximate primal and dual solutions

$$w^{(k+1)} = \frac{1}{\lambda n}\left(\rho^{k+2} p^{(k+1)} + q^{(k+1)}\right), \qquad x^{(k+1)} = \rho^{k+1} u^{(k+1)} + v^{(k+1)}.$$

---

Each iteration of Algorithm 5 only involves the two inner products $A_{i_k}^T p^{(k)}$, $A_{i_k}^T q^{(k)}$ in computing $\nabla_{i_k}^{(k)}$ and the two vector additions in (18). They all cost $O(d)$ rather than $O(n)$. When the $A_i$'s are sparse (the case of most large-scale problems), these operations can be carried out very efficiently. Basically, each iteration of Algorithm 5 only costs twice as much as that of SDCA [6, 19].

## 4 Experiments

In our experiments, we solve ERM problems with smoothed hinge loss for binary classification. That is, we pre-multiply each feature vector $A_i$ by its label $b_i \in \{\pm 1\}$ and use the loss function

$$\phi(a) = \begin{cases} 0 & \text{if } a \geq 1, \\ 1 - a - \frac{\gamma}{2} & \text{if } a \leq 1 - \gamma, \\ \frac{1}{2\gamma}(1-a)^2 & \text{otherwise.} \end{cases}$$

The conjugate function of $\phi$ is $\phi^*(b) = b + \frac{\gamma}{2}b^2$ if $b \in [-1, 0]$ and $\infty$ otherwise. Therefore we have

$$\Psi_i(x_i) = \frac{1}{n}\left(\phi^*(-x_i) - \frac{\gamma}{2}\|x_i\|^2\right) = \begin{cases} \frac{-x_i}{n} & \text{if } x_i \in [0, 1] \\ \infty & \text{otherwise.} \end{cases}$$

The dataset used in our experiments are summarized in Table 1.

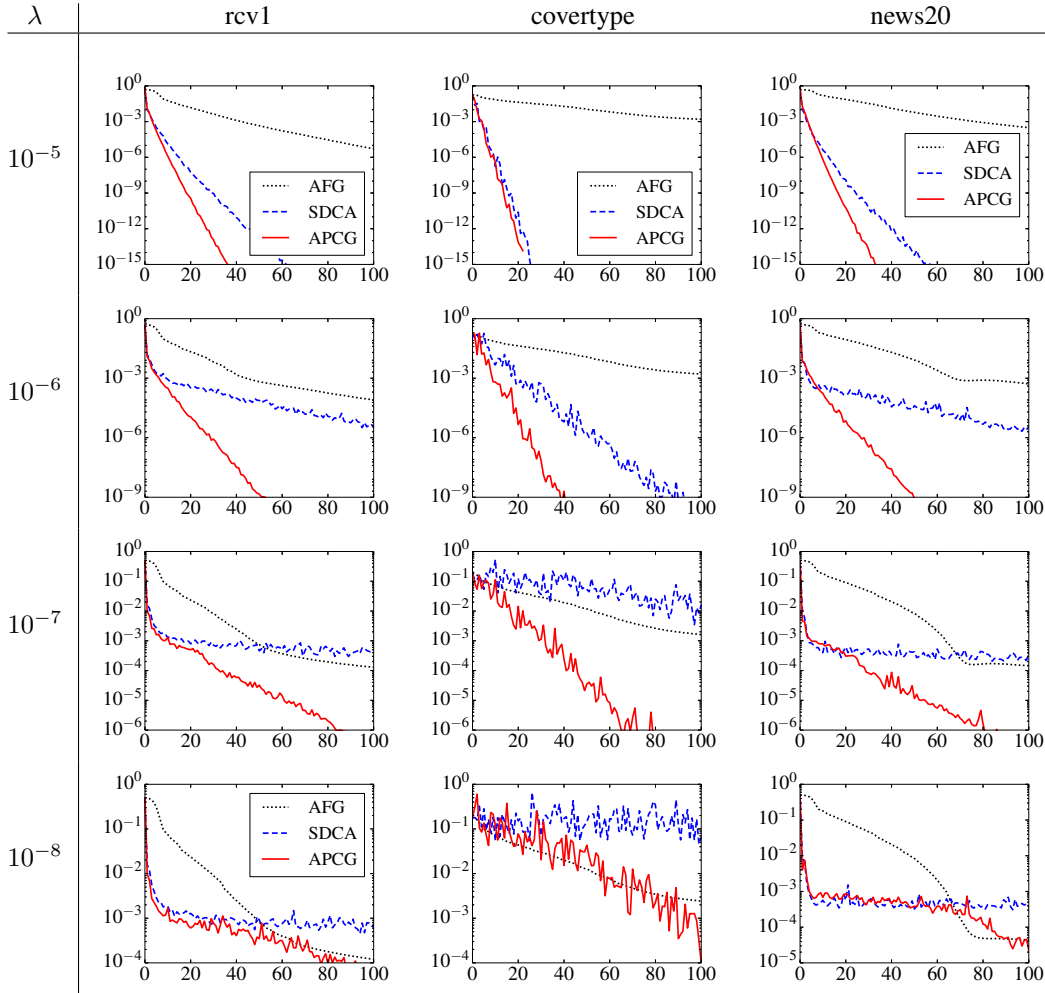

Figure 1: Comparing the APCG method with SDCA and the accelerated full gradient method (AFG) with adaptive line search. In each plot, the vertical axis is the primal objective gap $P(w^{(k)}) - P^\star$, and the horizontal axis is the number of passes through the entire dataset. The three columns correspond to the three datasets, and each row corresponds to a particular value of the regularization parameter $\lambda$.

In our experiments, we compare the APCG method with SDCA and the accelerated full gradient method (AFG) [12] with an additional line search procedure to improve efficiency. When the regularization parameter $\lambda$ is not too small (around $10^{-4}$), then APCG performs similarly as SDCA as predicted by our complexity results, and they both outperform AFG by a substantial margin.

Figure 1 shows the results in the ill-conditioned setting, with $\lambda$ varying form $10^{-5}$ to $10^{-8}$. Here we see that APCG has superior performance in reducing the primal objective value compared with SDCA and AFG, even though our theory only gives complexity for solving the dual ERM problem. AFG eventually catches up for cases with very large condition number (see the plots for $\lambda = 10^{-8}$).

| datasets | number of samples $n$ | number of features $d$ | sparsity |
|---|---|---|---|
| rcv1 | 20,242 | 47,236 | 0.16% |
| covtype | 581,012 | 54 | 22% |
| news20 | 19,996 | 1,355,191 | 0.04% |

Table 1: Characteristics of three binary classification datasets (available from the LIBSVM web page: `http://www.csie.ntu.edu.tw/~cjlin/libsvmtools/datasets`).

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
