[Reviews · NeurIPS 2014]

Submitted by Assigned_Reviewer_7

The authors propose an accelerated proximal block coordinate descent
algorithm, describe its application to standard regularized loss
minimization problems, and conclude with experiments on a smoothed SVM.

On the question of clarity: I found the paper on the whole difficult to
follow, with the authors showing a marked preference for writing
equations in lieu of explanations. There are also numerous small
grammatical errors.

I'm not aware of other algorithms that are designed to work on
block-coordinate problems (although single-coordinate algorithms are
common enough), and have to question the advantage of this formulation,
aside from being slightly more general. Given that the application
considered in section 4 is single-coordinate (am I correct about this?),
it might simplify the presentation to work from a single-coordinate
formulation, and merely mention that block-coordinate updates are also
possible.

On originality: I'm not familiar enough with the literature to judge. The
stated improvement over the best-known convergence rates is small (not
including the arxiv paper cited as [18]), and I think the authors
shouldn't minimize the difference between converging quickly in the
primal (as the other algorithms mentioned on the bottom of page 6 do) and
in the dual (as APCG, at least for the regularized ERM application). At
least for SVMs, with which I am most familiar, the two objectives are
very different, and fast convergence in the dual does not necessarily
imply fast convergence in the primal.

The experiments appear to show a clear win for APCG over SDCA, and I
agree with the authors that SDCA is a natural baseline. However, I do
have a question about a potential issue: in some of the plots,
particularly on the news20 dataset with small lambda, it appears that
SDCA "levels out" at a higher suboptimality than that found by APCG. This
indicates to me that perhaps at that point SDCA has converged, and has
done so to a slightly higher objective function value due to numerical
issues or implementation differences (the log scale exaggerating such
small discrepancies). Do you have any evidence that this is not the case?
For example, if you can show that your algorithm converges faster than
SDCA in terms of testing classification accuracy, it would simultaneously
assuage fears about a potential numerical issue, and demonstrate that the
use of APCG has practical benefits.

Also did you forget a square root in the equation on lines 190-191?
Summary: A difficult-to-read paper which describes an block coordinate descent
algorithm with a slightly better convergence rate than
previously-published (non-arXiv) work in the single-coordinate setting,
and experiments which may show a clear win over SDCA, although I have
some doubts (mentioned above).

Submitted by Assigned_Reviewer_23

The paper studies a new accelerated coordinate descent scheme, using proximal steps on the coordinates. This type of setting is highly relevant in machine learning currently, since this class of methods only requires access to one data example (or coordinate) per update step, as in SGD, see e.g. [19]. The paper is clearly and carefully written, and introduces novel proof techniques. In particular in comparison to the recent accelerated SDCA [18], the method here obtains the same fast and online rate, but is considerably simpler and avoids a tedious inner-outer iteration scheme, which is very attractive.
Compared to the recent APPROX method [4] (which is obtained as a special case), the main advantage here is that the method gives accelerated linear rates with automatic adaption to strongly convex objectives.

The presented experiments are convincing in my opinion, with the method showing advantages over SDCA even early at moderate precision, not only in the late high precision regime as with other accelerated methods. It would be nice to get more insights into the conditioning numbers and the dependence of the speedup on this number, when compared to SDCA.

It should be made more clear that the current theory only covers smooth loss. Can you comment on the extension to non-smooth convex losses?
(Added after the author feedback: While it's clear that the rates for the non-smooth case will be sub-linear, it is still important to comment on the comparison of these e.g. with the SDCA ones, no matter if using smoothing or not.)

Furthermore, it would be nice to comment about the case when the strong convexity parameter mu is not known.
(Thanks for answering, please make more clear in the algorithm description that the method can in general not be run in this case).

%%% minor comments:
l130: choose*s*
l297: maybe call it 'dual suboptimality' instead of 'dual optimality gap' (as it's not a gap)
same of the primal, e.g. in l411 etc, 'primal suboptimality' instead of 'primal objective value gap'
Summary: The paper studies a new accelerated coordinate descent scheme, using proximal steps on the coordinates. The main result being linear convergence under strongly convex objectives is an unexpected and welcome novelty for coordinate descent methods, improving over the known methods such as SDCA [19].

Submitted by Assigned_Reviewer_40

In this paper, the authors propose an accelerated algorithm for randomized proximal coordinate gradient method. Compared with previous work, their algorithm achieves a faster convergence rate. When the objective function is strongly convex, the proposed algorithm has a linear convergence rate and only depends on the square root of the condition number.

Besides the nice theoretical guarantees, the authors also considered the efficiency issues. In particular, for the empirical risk minimization problem, the cost of each iteration is independent of $n$.

After the SDCA algorithm is introduced, it is natural to ask whether it is possible to further accelerate SDCA. This paper provided an affirmative answer. The algorithm proposed in this paper is novel, and the paper is well-written.

Minor Points:
In the equation on Line 191, is a square root over $n \alpha_0$ missed?
It is better to discuss [18] in the introduction section.
A citation should be given for the acceleration technique in Algorithm 1, Section 2.
Summary: The authors propose a novel algorithm to accelerate stochastic coordinate gradient method, and provide solid theoretical guarantee. This paper made a significant contribution to composite convex optimization.
Author Feedback
Author rebuttal: First, we thank the reviewers for their comments and suggestions. In the following, we address their comments separately.

Reviewer_23:

In application to empirical risk minimization (ERM), we only give results for smooth loss. For non-smooth loss, the dual function -D(x) may not be strongly convex. We could directly apply Theorem 1 for the non-strongly convex case, but would get an inferior complexity. The better way is to add a small strongly convex term, say (\epsilon/2)||x||^2, to the dual -D(x), which corresponds to smoothing the primal loss function. Then we can obtain a complexity like in equation (16) with \gamma replaced by \epsilon. This also improves upon SDCA [19] for the case of non-smooth loss functions.

When the strong convexity parameter \mu is unknown, then we cannot obtain accelerated linear rates directly. Recently there are techniques developed to overcome this difficulty for the batch case (for example restart), and it is possible to extend them to APCG.

Reviewer_40:

For the equation on line 191, we should replace the first $n\alpha_0$ (in denominator) by $n\alpha_{-1}$, which equals $\sqrt{\gamma_0}$. The second $n\alpha_0$ should be $(n\alpha_{-1})^2$, which equals $\gamma_0$. We will correct it.

We will discuss [18] in the introduction, and add a citation to Nesterov for the acceleration technique used in Algorithm 1.

Reviewer_7:

We believe that our contributions in this paper are novel and significant. The construction and complexity analysis of accelerated coordinate gradient methods for composite convex optimization has been an open problem in the optimization community for a few years, with some recent progress made by Nesterov[12] and Fercoq and Richtarik[4]. For the first time our paper give the complete solution for both the strongly convex and non-strongly convex case. And the application to regularized ERM directly leads to improved complexity than SDCA. In addition to the theoretical significance, our experiments clearly demonstrate the superior performance of APCG on real datasets.

The block coordinate formulation is more general than single coordinate. For example, our results in Theorem 1 covers the accelerated full gradient methods as the special case of a full block, thus establishes a full spectrum of algorithms and their complexities. Moreover, block coordinate methods in the dual correspond to mini-batch stochastic gradient methods in the primal.

At the bottom of page 6, we discussed that our complexity is for the dual gap D*-D(x), while SDCA is for primal-dual gap P(w)-D(x), and SAG is for primal gap P(x)-P*. It is true that fast convergence in the dual does not imply fast convergence in primal. But our experiments in Section 4 clearly illustrate that APCG has superior performance in reducing the primal gap. In fact, in our recent work, we have also established same theoretical guarantee on reducing the primal gap by a simple variant of APCG, which will appear in a full length report.

For the experiments, we are confident on the behavior of SDCA on the news20 dataset, and have numerical evidence that it is not converging to a different value. Actually one can show that when \lambda is very small (<=10^{-7} in our experiments), then the updating rule of SDCA is extremely close to stochastic gradient method with certain constant stepsize, and it is expected to demonstrate the slow convergence behavior as we observe.

We will also polish the paper and fix small grammatical errors.